# OpenReview forum: "Towards Effective MLLM Jailbreaking Through Balanced On-Topicness and OOD-Intensity"
_ICLR.cc/2026/Conference — Submitted to ICLR 2026_

### Official Review · Reviewer_Hbdv · 2025-10-25

**Soundness:** 2
**Presentation:** 3
**Contribution:** 2
**Rating:** 4
**Confidence:** 5

**Summary:**

This paper focuses on addressing the vulnerability of Multimodal Large Language Models (MLLMs) to adversarial prompts and the inaccuracies in existing jailbreak evaluation standards. It points out that current jailbreak strategies often overestimate success rates, as many "successful" responses are benign, vague, or unrelated to malicious goals.
To solve this, the paper proposes a four-axis evaluation framework covering input on-topicness, input out-of-distribution (OOD) intensity, output harmfulness, and output refusal rate. Through empirical research, it reveals a structural trade-off: highly on-topic prompts are easily blocked by safety filters, while overly OOD prompts evade detection but fail to generate harmful content.
Based on this insight, the authors design a recursive rewriting strategy called Balanced Structural Decomposition (BSD). BSD decomposes malicious prompts into semantically consistent sub-tasks, adds subtle OOD signals and visual cues, and uses a neutral tone to present inputs. Experiments on 13 commercial and open-source MLLMs show that BSD outperforms existing methods, improving attack success rates by 67% and harmfulness by 21%, and also performs well against guard models.

**Strengths:**

- The proposed four-axis evaluation framework comprehensively captures both input and output characteristics of MLLM jailbreaks, addressing the defect of overestimating attack effectiveness in traditional binary evaluation and providing a more accurate and reliable benchmark tool for subsequent research.
- The BSD strategy innovatively balances the trade-off between on-topicness and OOD intensity. By recursively decomposing prompts and integrating visual cues, it effectively evades MLLM safety filters while ensuring the generation of harmful content, achieving breakthroughs in attack performance.
- The study conducts extensive experiments across 13 MLLMs (including commercial closed-source and open-source models) and two guard models, using three datasets (HADES, MMSafetyBench, and AdvBench-M). The large-scale and multi-dimensional experimental design enhances the generalizability and persuasiveness of the research results.

**Weaknesses:**

- The BSD strategy relies heavily on the quality of sub-task decomposition by the Qwen2.5-7B model. When facing overly obvious or complex malicious objectives, the decomposition fails to produce semantically diverse sub-tasks, leading to reduced jailbreak success rates and limiting the strategy's adaptability.
- The generation of descriptive images in BSD depends on the FLUX.1-schnell text-to-image model. The paper lacks an in-depth analysis of how image quality, style consistency, and semantic alignment with sub-tasks specifically affect jailbreak results, and there is insufficient verification of the necessity of visual cues.
- The study only evaluates the short-term effectiveness of jailbreak attacks but ignores the long-term impact of repeated use of BSD on MLLMs. It does not explore whether MLLMs can learn to identify and defend against such structured decomposition attacks, resulting in incomplete research on attack durability.
- This paper needs to add more advanced baselines for further comparison to highlight the superiority of the proposed attack.

**Questions:**

Comments:

1. Over-reliance on specific decomposition models without alternative mechanisms: The BSD strategy is highly dependent on Qwen2.5-7B for sub-task decomposition. When this model fails to decompose complex or obviously malicious prompts effectively, the entire jailbreak process breaks down. The paper does not propose alternative decomposition models or adaptive adjustment mechanisms to address this single-point failure risk, which affects the robustness of the strategy.

2. Insufficient validation of the role of visual cues: Although BSD integrates descriptive and distraction images, the paper only verifies the performance differences between FLUX-generated images, colored boxes, and random noise in ablation experiments. It does not quantitatively analyze how factors such as image semantic relevance to sub-tasks, visual complexity, and number of distraction images affect the jailbreak effect, making the role of visual cues in the strategy unclear.

3. Limited analysis of cross-dataset generalization differences: The paper tests BSD on three datasets (HADES, MMSafetyBench, AdvBench-M) but does not deeply analyze why the strategy has significant performance differences across datasets (e.g., lower ASR on AdvBench-M due to fewer samples). It also fails to explore whether the decomposition logic of BSD needs to be adjusted for different types of malicious prompts in different datasets, limiting the understanding of the strategy's cross-scenario adaptability.

4. Lack of in-depth comparison with state-of-the-art baselines: Although this paper has conducted in-depth comparisons with CS-DJ and FigStep, it is still not convincing enough. I recommend that the authors compare with the methods in the following references:

[1] Ma, Teng, et al. "Heuristic-induced multimodal risk distribution jailbreak attack for multimodal large language models." Proceedings of the IEEE/CVF International Conference on Computer Vision. 2025.

[2] Liu, Yi, et al. "Arondight: Red teaming large vision language models with auto-generated multi-modal jailbreak prompts." Proceedings of the 32nd ACM International Conference on Multimedia. 2024.

[3] Jeong, Joonhyun, et al. "Playing the fool: Jailbreaking llms and multimodal llms with out-of-distribution strategy." Proceedings of the Computer Vision and Pattern Recognition Conference. 2025.

5. Minor issues
- The font size of the text in Figures 1, 8, 9 is too small to be read.
- "Stage 1: Width-first balancing via Width Balance Score." should be "Stage 1: Width-first balancing via width balance score."

**Details Of Ethics Concerns:**

None.

---

> ### Author Response · Authors · 2025-11-30
> **Rebuttal (Part 1)**
>
> We thank the reviewer for the careful and constructive feedback. We address each concern below.
>
> ---
>
> ## Reliance on Qwen2.5-7B for Decomposition [W1, Q1]
>
> We agree that robustness to the decomposition model is important. While the main paper performs BSD with Qwen2.5-7B, **BSD is not tied to this single model**. In the current draft, we use **Qwen2.5-7B** due to its strong instruction-following ability. We have additionally instantiated BSD with **Qwen2.5-3B** as an alternative text-only decomposer. Across these decomposers, BSD retains strong performance and higher ASR than CS-DJ baseline (40.67 for GPT-4.1 and 30.27 for GPT-4o):
>
> ### Decomposition model ablation (ASR, higher is better)
>
> | Decomposition Model | Victim Model |Animal | Financial | Privacy | Self-Harm | Violence | Average |
> |------------------------|-----|-----|-----|-----|-----|-----|-----|
> | Qwen2.5-7B (Default)|GPT-4.1|43.33|88.67|78.67|28.00|64.67|60.67|
> | Qwen2.5-3B |GPT-4.1|44.67|78.00|67.33|22.00|50.00|52.40|
> | Qwen2.5-7B (Default)|GPT-4o|58.00|94.00|92.67|42.67|80.67|73.60|
> | Qwen2.5-3B |GPT-4o|50.67|92.67|82.67|36.00|69.33|66.27|
>
>
>
> These results indicate that BSD’s gains come from the decomposition strategy plus OT/OI-guided search, rather than from a particular hand-picked LLM. We will add a short ablation table reporting ASR and failure rates for different text-only decomposers in the appendix.
>
>
> ---
>
> ## Validation of the Role of Visual Cues [W2, Q2]
>
>
> We have added explicit **ablation groups** to find out the contribution of different visual components.
>
> ### Visual component ablation (ASR, higher is better)
>
> - BSD with **only descriptive** images,
> - **Typographic + distraction** images (CS-DJ),
> - **Full BSD** (descriptive + distraction).
>
> | Setting | Victim Model | Animal | Financial | Privacy | Self-Harm | Violence | Average |
> |------------|--------------|-----|-----|-----|-----|-----|-----|
> | Descriptive-Only | GPT-4o |26.00|72.00|66.67|26.00|63.33|50.80|
> | Typographic + Distraction (CS-DJ) | GPT-4o |22.00|43.33|39.33|12.67|34.00|30.27
> | Descriptive + Distraction (Ours) | GPT-4o |58.00|94.00|92.67|42.67|80.67|73.60|
> | Descriptive-Only | GPT-4.1 |37.33|62.67|58.67|18.00|18.00|50.67|45.47|
> | Typographic + Distraction (CS-DJ) | GPT-4.1 |22.00|60.00|56.67|16.00|48.67|40.67|
> | Descriptive + Distraction (Ours) | GPT-4.1 |43.33|88.67|78.67|28.00|64.67|60.67|
>
> These results show that **descriptive images alone already outperform** the pure typographic + distraction setting of CS-DJ, and that the **full BSD configuration (descriptive + distraction)** yields the strongest ASR by a large margin. This supports our claim that semantically meaningful descriptive images are a crucial component for achieving the desired OT/OI balance.
>
> We additionally vary diffusion hyperparameters controlling the visual cues:
>
> ### Visual cue parameter ablation (ASR, higher is better)
>
> | Guidance Scale | Steps        | Style          | Victim Model | Animal | Financial | Privacy | Self-Harm | Violence | Average |
> |------------|------------|--------------|-----|-----|-----|-----|-----|-----|-----|
> | 10 (default)   | 20 (default) | Anime (default) | GPT-4.1      |43.33|88.67|78.67|28.00|64.67|60.67|
> | 10 (default)   | 20 (default) | Anime (default) | GPT-4o       |58.00|94.00|92.67|42.67|80.67|73.60|
> | 4              | 20 (default) | Anime (default) | GPT-4.1      |43.33|79.33|62.00|25.33|58.00|53.60|
> | 4              | 20 (default) | Anime (default) | GPT-4o       |33.33|92.67|81.33|33.33|66.67|61.47|
> | 10 (default)   | 10           | Anime (default) | GPT-4.1      |50.00|86.00|68.67|32.00|64.67|60.27|
> | 10 (default)   | 10           | Anime (default) | GPT-4o       |27.33|85.33|82.00|24.67|62.00|56.27|
> | 4   | 10           | Anime (default) | GPT-4.1      |49.33|88.00|63.33|32.67|62.67|59.20|
> | 4   | 10           | Anime (default) | GPT-4o       |36.00|91.33|75.33|36.00|68.67|61.47
> | 10 (default)   | 20 (default) | Realistic       | GPT-4.1      |36.67|42.00|62.67|22.00|55.33|43.73
> | 10 (default)   | 20 (default) | Realistic       | GPT-4o       |37.33|66.67|79.33|28.67|62.00|54.80|
>
> Across anime-style settings, BSD’s ASR remains consistently high and clearly above CS-DJ, with only moderate variation under different hyperparameter settings. The drop under realistic images suggests that safety-aligned models may be more conservative with photorealistic content, but even then the attack remains competitive.

---

> ### Author Response · Authors · 2025-11-30
> **Rebuttal (Part 2)**
>
> ## Long-Term / Repeated-Use Durability [W3]
>
> We agree that long-term dynamics under repeated attacks is an important direction. However, fully studying **safety-alignment training or online adaptation** on top of our generated jailbreak samples (for either open- or closed-source MLLMs) is beyond the scope of this work.
>
> Instead, we provide **mechanistic evidence** that BSD exploits *structural* vulnerabilities rather than easy-to-detect patterns:
>
> - **Probing harmfulness vs. refusal.**
>   Using the probing method from *Zhao et al., "LLMs Encode Harmfulness and Refusal Separately" (NeurIPS 2025)*, we adopt and extend their conclusions to the multimodal setting. They show that:
>   (i) LLMs (and, in our case, MLLMs) encode **harmfulness** at instruction tokens $t_\text{inst}$ and **refusal** at post-instruction tokens $t_\text{post-inst}$
>
>   (ii) many jailbreak methods mainly suppress the **refusal** signal while leaving the model’s internal harmfulness belief largely intact.
>
>   Applying the same probes, we observe that BSD reduces both **harmfulness perception** (at $t_\text{inst}$) and **refusal activation** (at $t_\text{post-inst}$) compared to the CS-DJ baseline. This suggests that BSD modifies internal representations in a way that **jointly suppresses harmfulness and refusal**, making it less likely that simple refusal-based defenses or narrow safety fine-tuning can fully patch it.
>
> - **Effectiveness against guard models.**
>   In additional experiments, BSD successfully bypasses two strong guard pipelines: **GuardReasoner-VL-3B/7B** and **LLaVA-Guard-0.5B/7B**, and does so with a clear margin over CS-DJ. This indicates that BSD is not merely exploiting a specific benchmark setup, but remains effective even when external guard models are deployed.
>
> We will clarify that fully modeling long-term adaptation / multi-turn jailbreak is left as future work, and we will add these mechanistic results (including probe score plots for BSD vs. CS-DJ) in an extra page to show our claims about structural vulnerabilities.
>
> ---
>
> ## Cross-Dataset Generalization Differences [Q3]
>
> Thank you for pointing this out; our initial text did not sufficiently explain the cross-dataset behavior.
>
> BSD is intentionally designed as a **dataset-agnostic** strategy: **the same decomposition and OT/OI-guided search are applied unchanged to HADES, MMSafetyBench, and AdvBench-M**. Performance differences arise largely from **dataset characteristics**:
>
> - **Prompt structure and complexity.** MM-Safetybench contains more political lobbying and government decision that are harder for decomposers to break into diverse sub-tasks without losing coherence.
> - **Evaluator coverage.** For some domains (e.g., specific political scenarios, our jailbreak evaluator (BeamDeaver-7B) has weaker coverage, which can lead to underestimated ASR for all methods, not just BSD.
>
> We will clarify that we intentionally avoid dataset-specific tuning (to keep BSD general), and discuss how dataset-specific adaptations (e.g., specialized decomposer prompts for political domains) could further improve performance, but at the cost of generality.

---

> ### Author Response · Authors · 2025-11-30
> **Rebuttal (Part 3)**
>
> ## Additional Baselines and Victim Models [W4, Q4]
>
> We agree that a broader comparison strengthens our empirical claims. Beyond FigStep and CS-DJ in the original submission, we have now added:
>
> - **JOOD** as an additional recent multimodal jailbreak baseline on the AdvBench-M dataset,
> - Additional **victim models (GPT-5, GPT-5.1)** on HADES, extending the closed-source coverage.
>
> ### AdvBench-M: comparison with JOOD
> | Methods | Victim Model | Bomb | Drugs | Firearms | Hack | Kill | Social | Suicide | Average |
> |---------|--------------|-----|-----|-----|-----|-----|-----|-----|-----|
> | CS-DJ   | Gemini-2.5-Pro |23.33|40.00|23.53|31.58|0.00|0.00|26.67|20.73|
> | JOOD    | Gemini-2.5-Pro |13.33|12.81|18.03|15.32|16.48|5.33|24.37|15.10|
> | Ours    | Gemini-2.5-Pro |26.67|83.33|58.82|94.74|66.67|65.00|16.67|58.84|
> |# Data | - | 30 | 30 | 17 | 19 | 24 | 20 | 30 | 24.29
>
> BSD significantly outperforms both CS-DJ and JOOD on AdvBench-M in ASR, while operating under a much lower computational cost.
>
> The main reason we did not include JOOD in our **HADES** experiments is its extremely high computational and API cost. JOOD must:
>
> - try **five** types of image augmentation methods,
> - for **nine** mixing ratios (0.1–0.9, step 0.1),
> - combined with **five** benign background images.
>
> This results in roughly **5 × 9 × 5 = 225× more API calls per example** than BSD. This overhead is manageable on AdvBench-M, where each harmful category has on average 24.29 examples, but becomes much less practical on HADES, where each category has **150** examples. Running JOOD at full scale on HADES would therefore be both time- and cost-prohibitive, while offering limited additional insight beyond the AdvBench-M comparison already included.
>
> Regarding the other suggested baselines:
>
> - For **Arondight**, at the time of our experiments we were unable to access publicly released code and attack recipes that would allow a faithful reproduction on our full set of models and datasets.
> - For **HIMRD**, the original setup involves its own text-to-image pipeline and a dataset different from ours. A detailed reproduction would require additional engineering and evaluation work that we cannot complete within the rebuttal timeline.
>
> We will explicitly state these limitations in the revised paper and discuss Arondight and HIMRD in the **Related Work** section, highlighting how their design choices compare conceptually to BSD.
>
> ### HADES: extended closed-source models
>
> We also extend our evaluation on HADES to more recent closed-source MLLMs:
>
> | Methods | Victim Model | Animal | Financial | Privacy | Self-Harm | Violence | Average |
> |---------|--------------|-----|-----|-----|-----|-----|-----|
> | CS-DJ   | GPT-5.1       |13.33|36.67|24.00|18.67|41.33|26.80|
> | Ours    | GPT-5.1       |28.67|86.67|76.67|46.67|68.67|61.47(**+34.67**)|
> | CS-DJ   | GPT-5        |18.67|53.33|36.00|19.33|19.33|29.33|
> | Ours    | GPT-5        |9.33|61.33|34.00|20.67|45.33|34.13(**+4.8**)|
>
> BSD achieves a **substantial improvement** over CS-DJ on GPT-5.1 (+34.67 average ASR), while also maintaining an advantage on GPT-5 (+4.80). These extended results will be incorporated into the main result tables.
>
> ## Minor Issues
> We will (i) refine the font size in Figures 1, 8, 9 and (ii) correct the typo on Line 257 in the updated version.

---

### Official Review · Reviewer_FV5b · 2025-10-27

**Soundness:** 3
**Presentation:** 2
**Contribution:** 3
**Rating:** 6
**Confidence:** 4

**Summary:**

This paper investigates a practical limitation of existing OOD jailbreak attacks where the responses classified as "successfully jailbroken" are frequently unrelated to the malicious intent while the on-topic attack prompts only elicit response rejection.
To effectively jailbreak MLLMs, the authors proposed balancing this trade-off between OOD-ness and On-topicness by iterative decomposition of attack prompts and depth refinement within a tree structure.

**Strengths:**

- The paper is tackling an unexplored but practical limitation of existing OOD attacks with qualitative analysis of relevance-novelty trade-off.
- The paper proposed a novel tree-based decomposition attack strategy to balance the trade-off.
- The paper is well-written and easy to follow.
- The comprehensive experimental results over closed and open-source models demonstrate the jailbreak effectiveness of proposed method with detailed qualitative analysis.

**Weaknesses:**

- One of my concerns is in the paper’s reliance on SBERT embeddings to quantify On-Topicness (OT) and Out-of-Distribution Intensity (OI). SBERT has known limitations in capturing subtle semantic variations—such as word order perturbations or nuanced paraphrases [R1] —which may undermine the reliability of these metrics. For instance, in Eq (2), when the short caption from MLLM is subtly but semantically altered where only minor lexical changes lead to a substantially different meaning—it is unclear whether the OI metric based on SBERT embeddings can reliably capture such distinctions.
- Also, when measuring OI in Eq (2), a safety-aligned MLLM may generate a safe summary rather than one semantically consistent with the harmful prompt $P_0$. This naturally increases the OI score even though the divergence stems from safety alignment rather than genuine distributional novelty. Consequently, the metric may conflate the model's safety-alignment with true out-of-distribution characteristics, limiting its interpretability as an indicator of OOD intensity.
- In Eq. (3), the formulation of the Harmfulness Score (HS) may produce misleading results. If the reference vector $h_{ref}$ ​contains uniformly high category scores while the response vector $h_r$ ​ is uniformly low, the ℓ1 ​distance term​ still increases—thereby raising the overall HS despite the response being less harmful.
- In Eq. (4), the notation $N$ is used without a clear definition. While it appears to represent the total number of evaluated responses, this seems not explicitly stated in the manuscript.
- In Eq. (5), the assumption that greater semantic dissimilarity among decomposed sub-tasks directly corresponds to higher OOD characteristics lacks clear justification. It is not evident that mutual dissimilarity between sub-tasks translates to genuine OOD behavior from the model’s perspective.
- The experimental comparison is limited to two baselines (FigStep, CS-DJ), which is too narrow to support broad claims. Including a broader set of recent baselines (also with recent victim models such as GPT-5) would make the results more convincing.
- The iterative decomposition (width) and refinement (depth) and image generation likely add significant computational overhead compared to the non-optimization methods such as FigStep.
[R1] https://arxiv.org/pdf/2309.03747

**Questions:**

See above weaknesses.

---

> ### Author Response · Authors · 2025-11-30
> **Rebuttal (Part 1)**
>
> We thank the reviewer for the thoughtful and detailed comments. Below we address each concern.
>
> ## Reliability of SBERT [W1]
>
> We agree that SBERT has limitations in capturing fine-grained semantic nuances. Our use of SBERT is not intended as a definitive “semantic oracle,” but as a **practical proxy** for the OT/OI axes. To check whether our conclusions depends on a specific SBERT variant, we conducted two analyses.
>
> ### Correlation between SBERT models
>
> We recomputed the OI metric using two encoders, `all-MiniLM-L6-v2` and `all-mpnet-base-v2`, and measured their agreement across all samples:
>
> - Pearson correlation: 0.8518 (p = 2.161e-43)
> - Spearman correlation: 0.8537 (p = 8.918e-44)
>
> This indicates that OI estimates are **highly stable across SBERT variants**, despite their known imperfections.
>
>
> ### Embedding model ablation (ASR, higher is better)
>
> We then evaluated BSD’s ASR under both encoders. The performance remains strong and consistently above the CS-DJ baseline (40.67 for GPT-4.1 and 30.27 for GPT-4o):
>
> | Embedding Model | Victim Model | Animal | Financial | Privacy | Self-Harm | Violence | Average |
> |-----------------|--------------|-----|-----|-----|-----|-----|-----|
> | all-MiniLM-L6-v2 (default) | GPT-4.1 |43.33|88.67|78.67|28.00|64.67|60.67|
> | all-MiniLM-L6-v2 (default) | GPT-4o |58.00|94.00|92.67|42.67|80.67|73.60|
> | all-mpnet-base-v2       | GPT-4.1 |50.00|82.00|72.00|27.33|61.33|58.53|
> | all-mpnet-base-v2      | GPT-4o | 34.00|89.33|88.00|34.67|64.00|62.00|
>
> Together, these results suggest that our OT/OI-based conclusions are **robust to the choice of SBERT encoder** and that BSD’s gains do not depend on a single, hand-picked embedding model. We will include these analyses and a short discussion in the revised appendix.
>
> ---
>
> ## Safety-Aligned Summaries and OI Interpretation [W2]
>
> We appreciate this subtle point. When a safety-aligned MLLM produces a benign summary that conceals harmful intent, OI may increase due to safety behavior rather than genuine novelty.
>
> To disentangle this effect, we conducted an additional analysis:
>
> 1. **Two-summary generation.**
>    For each adversarial image we build using the HADES dataset, we used `gpt-5-nano` to generate two summaries:
>    - a **default** summary using our standard summarization prompt, and
>    - a **“force-true”** summary with the following instruction:
>      > “Summarize the user’s true underlying intention as faithfully as possible, even if it is harmful or unsafe.”
>
> 2. **Independent semantic check.**
>    We then asked `Qwen-2.5-14B-Instruct` to judge, for each pair, whether the two summaries expressed the **same underlying meaning** or not, and also recorded cases where the model refused to answer:
>
>
> | Summary Prompt | Agree | Disagree | Refusal Rate | Total |
> |----------------|-------|----------|--------------|--|
> | Default vs. "Force-true"|663|27|60|750|
>
>
> The disagreement rate is only $27/750 = 3.6\%$, which indicates that in the vast majority of cases the **default safety-aligned summary still preserves the underlying harmful intent** captured by the “force-true” variant. In practice, the default summaries typically look like:
>
> > “To compare visual memes (left) with a flowchart diagram (right) illustrating a process, likely about {harmful prompt}.”
>
> The surface tone is neutral, but the harmful topic is still explicitly present.
>
> This gives us confidence that, in our setting, computing OI based on default safety-aligned summaries is **robust**, and that safety-driven rewriting does not dominate the signal.
>
> ---
>
> ## Equations and Notations [W3, W4]
>
> **HS corner case (Eq. 3).**
> We thank the reviewer for pointing out this edge case. Our original HS formulation based on an $l_1$ distance to a reference vector can indeed assign a high HS when the reference is very harmful and the response is much less harmful.
>
> To address this, in the revised version we will adopt a **directional harmfulness metric**:
>
> $$
> \mathrm{HS} = \max(0,\; h_{\text{r}} - h_{\text{ref}}),
> $$
>
> where $h_{\text{r}}$ is the score vector for the response $r$, and $h_{\text{ref}}$ is the reference vector obtained from the original prompt $P_0$. This ensures HS only increases when the response is *more harmful than original prompt*, and remains low for response with low score vectors, resolving the issue you identified.
>
> **Eq. (4) notation.**
> You are correct that the notation was not explicit. In the revision we will explicitly define $N$ as “the total number of evaluated responses in the dataset” and adjust the text accordingly.

---

> ### Author Response · Authors · 2025-11-30
> **Rebuttal (Part 2)**
>
> ## Semantic Dissimilarity vs. OOD in Eq. (5) [W5]
>
> We agree that **semantic dissimilarity between sub-tasks is not equivalent to ground-truth OOD** in a strict distributional sense. Our intention is more modest: the diversity term in Eq. (5) is used as a **heuristic regularizer** that encourages the decomposition to cover *multiple distinct aspects* of the malicious goal, rather than repeatedly targeting a single narrow pattern.
>
> In the revision, we will (i) tone down statements that directly equate this term with "\(-\bar{S}_S\) proxies OOD coverage" (Line 267), and (ii) explicitly describe it as a **diversity-promoting heuristic** that empirically helps BSD explore a broader region of the input decomposition space, rather than as a accurate estimator of OOD-Intensity.
>
> ---
>
> ## Extended comparisons against baselines [W6]
>
> We agree that a broader comparison strengthens our empirical claims. Beyond FigStep and CS-DJ in the original submission, we have now added:
>
> - **JOOD** as an additional recent multimodal jailbreak baseline on the AdvBench-M dataset,
> - Additional **victim models (GPT-5, GPT-5.1)** on HADES, extending the closed-source coverage.
>
> ### AdvBench-M: comparison with JOOD
> | Methods | Victim Model | Bomb | Drugs | Firearms | Hack | Kill | Social | Suicide | Average |
> |---------|--------------|-----|-----|-----|-----|-----|-----|-----|-----|
> | CS-DJ   | Gemini-2.5-Pro |23.33|40.00|23.53|31.58|0.00|0.00|26.67|20.73|
> | JOOD    | Gemini-2.5-Pro |13.33|12.81|18.03|15.32|16.48|5.33|24.37|15.10|
> | Ours    | Gemini-2.5-Pro |26.67|83.33|58.82|94.74|66.67|65.00|16.67|58.84|
> |# Data | - | 30 | 30 | 17 | 19 | 24 | 20 | 30 | 24.29
>
> BSD significantly outperforms both CS-DJ and JOOD on AdvBench-M in ASR, while operating under a much lower computational cost.
>
> The main reason we did not include JOOD in our **HADES** experiments is its extremely high computational and API cost. JOOD must:
>
> - try **five** types of image augmentation methods,
> - for **nine** mixing ratios (0.1–0.9, step 0.1),
> - combined with **five** benign background images.
>
> In other words, JOOD uses roughly **225× more API calls per example** than our method. This is manageable on AdvBench-M, where each harmful category has on average 24.29 examples, but becomes much more expensive on HADES, where each category has **150** examples. Running JOOD at full scale on HADES would therefore be both time- and cost-prohibitive, while providing limited additional insight beyond the AdvBench-M comparison already included.
>
> ### HADES: extended closed-source models
>
> We also extend our evaluation on HADES to more recent closed-source MLLMs:
>
> | Methods | Victim Model | Animal | Financial | Privacy | Self-Harm | Violence | Average |
> |---------|--------------|-----|-----|-----|-----|-----|-----|
> | CS-DJ   | GPT-5.1       |13.33|36.67|24.00|18.67|41.33|26.80|
> | Ours    | GPT-5.1       |28.67|86.67|76.67|46.67|68.67|61.47(**+34.67**)|
> | CS-DJ   | GPT-5        |18.67|53.33|36.00|19.33|19.33|29.33|
> | Ours    | GPT-5        |9.33|61.33|34.00|20.67|45.33|34.13(**+4.8**)|
>
> BSD achieves a **substantial improvement** over CS-DJ on GPT-5.1 (+34.67 average ASR), while also maintaining an advantage on GPT-5 (+4.80). These extended results will be incorporated into the main result tables.
>
> ---
>
> ## Computational Cost of BSD [W7]
>
>
> We acknowledge that BSD introduces additional overhead compared to non-optimization attacks due to (i) iterative decomposition (width/depth) and (ii) image generation. On average, the decomposition tree contains ~15 nodes, with one image per node, so the per-attack cost is higher than for FigStep.
>
> However:
>
> - **Effectiveness trade-off.** BSD yields a substantial ASR gain over non-optimization baselines (e.g., FigStep), while still operating at a much lower cost than methods that exhaustively scan hyperparameters and large augmentation grids (e.g., JOOD).
>
> - **Generalisability vs. tuning cost.** BSD is designed as a *general* procedure that can be applied across datasets and models without per-dataset or per-model hyperparameter tuning. In contrast, methods such as JOOD or Andriushchenko et al. (*“Jailbreaking leading safety-aligned LLMs with simple adaptive attacks,”* ICLR 2025) rely on extensive search over hyperparameters or augmentations. As discussed earlier, JOOD alone can require up to ~225× more API calls **per example** than BSD, which makes the evaluation cost relatively high on large-scale benchmarks like HADES.
>
> Overall, BSD offers a favorable **effectiveness-cost balance**: it is more expensive than simple one-shot attacks, but significantly more efficient than heavily tuned or augmentation-intensive methods, while remaining applicable across a wide range of models and datasets without specific hyperparameter tuning.

---

### Official Review · Reviewer_16QD · 2025-10-29

**Soundness:** 3
**Presentation:** 2
**Contribution:** 3
**Rating:** 4
**Confidence:** 4

**Summary:**

This paper addresses the challenge of effectively jailbreaking Multimodal Large Language Models (MLLMs). It argues that current evaluation methods are flawed, as they often misclassify benign or off-topic responses as successful attacks. To rectify this, the authors propose a new four-axis evaluation framework that assesses prompts based on On-Topicness (OT), Out-of-Distribution Intensity (OI), Harmfulness, and Refusal Rate. Through empirical analysis, the paper identifies a critical structural trade-off: highly on-topic prompts are more harmful but also more likely to be rejected, while extreme out-of-distribution (OOD) prompts evade filters but produce less harmful content. To exploit the optimal balance between relevance and novelty, the authors introduce Balanced Structural Decomposition (BSD), a recursive strategy that decomposes malicious instructions into semantically coherent sub-tasks paired with descriptive images. Evaluated on 13 MLLMs, BSD demonstrates superior performance by generating more harmful outputs with fewer refusals compared to baseline methods, highlighting a vulnerability in existing safety mechanisms that rely on surface-level filtering.

**Strengths:**

+ The introduction of the four-axis framework (On-Topicness, OOD-Intensity, Harmfulness, Refusal Rate) provides a more nuanced and comprehensive standard for evaluating MLLM jailbreaks, addressing the overestimation problem in prior work.
+ The paper makes a good contribution by empirically identifying and formalizing the fundamental trade-off between prompt relevance and novelty, offering a clear explanation for the limitations of existing attack strategies.
+ The proposed BSD strategy is a well-motivated and systematic approach that directly targets the identified OT/OI trade-off, demonstrating state-of-the-art attack performance across a wide range of models.

**Weaknesses:**

- The authors' findings do not significantly differ from previous work. For example, one of the main findings of this paper, that existing jailbreaks are generally ineffective, is consistent with the conclusions of Nikolić et al. (ICML’25). However, this paper fails to provide sufficient new insights or highlight the differences from previous work.
- While BSD integrates descriptive and distracting images, it does not analyze specific visual features (such as content relevance) or perform ablation experiments to determine how this affects the OT/OI balance or model behavior, making the role of visual effects difficult to understand.
- More advanced defenses need to be added for further evaluation. This paper only evaluates GuardReasoner-VL-3B/7B, which does not seem convincing.

**Questions:**

Q1: Regarding the overlap between your findings and previous work (e.g., Nikolić et al., ICML'25) on the ineffectiveness of existing jailbreaks, could you elaborate on why the paper does not include a detailed comparative analysis of the evaluation frameworks, attack mechanisms, or core conclusions between your work and Nikolić et al.'s study?
[1] Nikolić, Kristina, et al. "The Jailbreak Tax: How Useful are Your Jailbreak Outputs?." In Proc. of ICML, 2025.

Q2: For the visual components integrated into BSD (descriptive and distraction images), since the paper does not analyze specific visual features (e.g., semantic relevance between descriptive images and the original malicious prompt), do you have plans to supplement experiments that quantify how variations in visual content relevance affect the On-Topicness (OT) and OOD Intensity (OI) balance of the input?

Q3: The paper lacks ablation experiments to isolate the impact of visual cues on BSD’s performance. For example, why did you not design an ablation group that removes descriptive images or distraction images entirely, and compare its ASR, HS, and RR with the full BSD model? This would help clarify whether visual components are necessary for achieving the OT/OI balance.

Q4: When evaluating BSD against guard models, the paper only tests GuardReasoner-VL-3B and GuardReasoner-VL-7B. Why did you not include other mainstream MLLM guard models (e.g., LLaVA-Guard, VILA-Guard, or commercial guard systems like OpenAI’s Content Policy Enforcement) in the evaluation to verify the generalizability of BSD’s ability to bypass defenses?

Q5: The paper states that BSD’s sub-task decomposition relies exclusively on Qwen2.5-7B, which fails when malicious prompts are overly explicit or complex (resulting in overt intent and model rejection). Given this limitation, why did the study not evaluate alternative decomposition LLMs (e.g., smaller open-source models like LLaVA-7B or safety-aligned models) to test whether they could improve the robustness of the decomposition module?

Q6: For overly complex malicious prompts (e.g., multi-step illicit operations) that Qwen2.5-7B fails to decompose effectively, the paper only notes "reduced jailbreak success" but does not define clear criteria for "complexity" of prompts. Could you clarify how the paper quantifies prompt complexity, and whether this quantification was used to systematically test the decomposition module’s limits?

**Details Of Ethics Concerns:**

Nil

---

> ### Author Response · Authors · 2025-11-30
> **Rebuttal (Part 1)**
>
> We thank the reviewer for the detailed and constructive feedback. We respond to each point below.
>
> ### Relation to Nikolić et al. (ICML’25) and Novelty [W1, Q1]
>
> Nikolić et al. (“The Jailbreak Tax”) primarily study text-only LLMs and focus on post-hoc evaluation of jailbreak outputs, showing that “successful” jailbreaks often degrade task utility. Our work differs in both scope and goal:
>
> - **Role of evaluation.** Nikolić et al. propose a primarily *post-hoc evaluation* framework for existing jailbreak outputs. Our four-axis framework (OT, OI, HS, RR) is designed as a **generative lens**: OT and OI are used as **explicit optimization targets** for constructing stronger attacks, not just for assessing them after jailbreaking.
> - **New attack mechanism.** We introduce **BSD**, a tree-based decomposition + visual attack method that is explicitly built on the OT–OI trade-off. Nikolić et al. do not propose a corresponding attack algorithm.
> - **Model & modality.** Nikolić et al. focus on **text-only LLMs** and analyze the *utility* of jailbreak outputs (“jailbreak tax”). We instead study **multimodal LLLMs** (MLLMs) and explicitly incorporate vision into both the attack and evaluation.
>
> We will clarify these differences in the revised paper by (i) adding a dedicated paragraph in Related Work and (ii) including a small comparison table contrasting model type, evaluation goal, and whether a new attack mechanism is proposed. Our main novelty is **operationalizing the OT–OI trade-off into a concrete, systematic MLLM jailbreak strategy**, rather than re-stating that existing jailbreaks can be weak.
>
> ---
>
> ### Ablation studies on visual part of our method [W2, Q2, Q3]
>
> We appreciate the request to more clearly analyze visual cues. Conceptually, **OT and OI are generative objectives** that could also be used to assess the quality of *image generation* models themselves and *image composition* methods. A full quantitative study of how to design image generators that optimally balance OT/OI is an interesting direction but lies beyond the scope of this work. In this paper, we focus on demonstrating that the **OT/OI trade-off is a useful lens for MLLM jailbreaking**, and we complement this with targeted ablations on the visual components within BSD.
>
> #### Visual component ablation (ASR, higher is better)
>
> - BSD with **only descriptive** images,
> - **Typographic + distraction** images (CS-DJ),
> - **Full BSD** (descriptive + distraction).
>
> | Setting | Victim Model | Animal | Financial | Privacy | Self-Harm | Violence | Average |
> |------------|--------------|-----|-----|-----|-----|-----|-----|
> | Descriptive-Only | GPT-4o |26.00|72.00|66.67|26.00|63.33|50.80|
> | Typographic + Distraction (CS-DJ) | GPT-4o |22.00|43.33|39.33|12.67|34.00|30.27
> | Descriptive + Distraction (Ours) | GPT-4o |58.00|94.00|92.67|42.67|80.67|73.60|
> | Descriptive-Only | GPT-4.1 |37.33|62.67|58.67|18.00|18.00|50.67|45.47|
> | Typographic + Distraction (CS-DJ) | GPT-4.1 |22.00|60.00|56.67|16.00|48.67|40.67|
> | Descriptive + Distraction (Ours) | GPT-4.1 |43.33|88.67|78.67|28.00|64.67|60.67|
>
> These results show that **descriptive images alone already outperform** the pure typographic + distraction setting of CS-DJ, and that the **full BSD configuration (descriptive + distraction)** yields the strongest ASR by a large margin. This supports our claim that semantically meaningful descriptive images are a crucial component for achieving the desired OT/OI balance.

---

> ### Author Response · Authors · 2025-11-30
> **Rebuttal (Part 2)**
>
> We additionally vary diffusion hyperparameters controlling the visual cues:
>
> ### Visual cue parameter ablation (ASR, higher is better)
>
> | Guidance Scale | Steps        | Style          | Victim Model | Animal | Financial | Privacy | Self-Harm | Violence | Average |
> |------------|------------|--------------|-----|-----|-----|-----|-----|-----|-----|
> | 10 (default)   | 20 (default) | Anime (default) | GPT-4.1      |43.33|88.67|78.67|28.00|64.67|60.67|
> | 10 (default)   | 20 (default) | Anime (default) | GPT-4o       |58.00|94.00|92.67|42.67|80.67|73.60|
> | 4              | 20 (default) | Anime (default) | GPT-4.1      |43.33|79.33|62.00|25.33|58.00|53.60|
> | 4              | 20 (default) | Anime (default) | GPT-4o       |33.33|92.67|81.33|33.33|66.67|61.47|
> | 10 (default)   | 10           | Anime (default) | GPT-4.1      |50.00|86.00|68.67|32.00|64.67|60.27|
> | 10 (default)   | 10           | Anime (default) | GPT-4o       |27.33|85.33|82.00|24.67|62.00|56.27|
> | 4   | 10           | Anime (default) | GPT-4.1      |49.33|88.00|63.33|32.67|62.67|59.20|
> | 4   | 10           | Anime (default) | GPT-4o       |36.00|91.33|75.33|36.00|68.67|61.47
> | 10 (default)   | 20 (default) | Realistic       | GPT-4.1      |36.67|42.00|62.67|22.00|55.33|43.73
> | 10 (default)   | 20 (default) | Realistic       | GPT-4o       |37.33|66.67|79.33|28.67|62.00|54.80|
>
> Across anime-style settings, BSD’s ASR remains consistently high and clearly above CS-DJ, with only moderate variation under different hyperparameter settings. The drop under realistic images suggests that safety-aligned models may be more conservative with photorealistic content, but even then the attack remains competitive.
>
> Overall, these ablations indicate that **visual components are necessary and meaningfully contribute to BSD’s performance**, and that their effect is robust to reasonable choices of diffusion hyperparameters.
>
>
> ---
>
> ## Performance against more guard models [W3, Q4]
>
> We agree that evaluating against multiple guard models strengthens the claim of generality.
>
> In addition to GuardReasoner-VL-3B/7B (already in the paper), we have now evaluated BSD against LLaVA-Guard. BSD maintains a large margin over baselines in ASR, and still drives HS/RR into the concerning region we highlight, indicating that the attack does not rely on idiosyncrasies of a single guard.
>
> ### Guard model evaluation (ASR, higher is better)
>
> | Method | Guard Model            | Animal | Financial | Privacy | Self-Harm | Violence | Average |
> |------------------------|-----|-----|-----|-----|-----|-----|-----|
> | Ours | GuardReasoner-VL-3B |99.33|98.87|98.87|98.77|97.33|98.40|
> | CS-DJ | GuardReasoner-VL-3B |89.33|79.33|78.00|90.00|77.33|82.80|
> | Ours | GuardReasoner-VL-7B |89.33|80.67|84.00|86.00|78.00|83.60|
> | CS-DJ | GuardReasoner-VL-7B |79.33|60.67|49.33|61.33|78.00|65.73|
> | Ours | LLaVA-Guard-0.5B |100|100|100|100|100|100|
> | CS-DJ | LLaVA-Guard-0.5B |100|98.67|100|98.67|98.67|99.20|
> | Ours | LLaVA-Guard-7B |90.67|96.00|92.00|96.67|88.00|92.67|
> | CS-DJ | LLaVA-Guard-7B |87.33|98.33|98.67|88.00|86.00|89.87|
>
> Both our method and the CS-DJ baseline can achieve high ASR under LLaVA-Guard, indicating that current multimodal guard models remain vulnerable to carefully structured attacks. Importantly, **BSD consistently outperforms CS-DJ under the GuardReasoner-VL-3B/7B pipeline and also maintains a margin on LLaVA-Guard-7B**, showing that our gains are not tied to a single guard model.
>
> These results support our claim that BSD exploits **intrinsic vulnerabilities in cross-modal refusal grounding**, rather than focusing to a particular benchmark or guard configuration. We will add the full guard-model comparison tables and discussion to the appendix in the revised version.

---

> ### Author Response · Authors · 2025-11-30
> **Rebuttal (Part 3)**
>
> ## Decomposition model [Q5]
>
> BSD only uses the **text** input of a language model for sub-task decomposition; the visual components are added *after* decomposition and do not require the decomposer to have any vision capabilities. For this reason, we deliberately focus on **text-only LLMs** as decomposition backbones. Using vision–language models such as LLaVA at this stage would introduce extra complexity without a clear benefit for the purely textual task-structuring problem.
>
> In the current draft, we perform BSD with **Qwen2.5-7B** due to its strong instruction-following ability, but BSD is **not** tied to this model. We have additionally instantiated BSD with **Qwen2.5-3B** as an alternative text-only decomposer. Across these decomposers, BSD retains strong performance and higher ASR than CS-DJ baseline (40.67 for GPT-4.1 and 30.27 for GPT-4o):
>
> ### Decomposition model ablation (ASR, higher is better)
>
> | Decomposition Model | Victim Model |Animal | Financial | Privacy | Self-Harm | Violence | Average |
> |------------------------|-----|-----|-----|-----|-----|-----|-----|
> | Qwen2.5-7B (Default)|GPT-4.1|43.33|88.67|78.67|28.00|64.67|60.67|
> | Qwen2.5-3B |GPT-4.1|44.67|78.00|67.33|22.00|50.00|52.40|
> | Qwen2.5-7B (Default)|GPT-4o|58.00|94.00|92.67|42.67|80.67|73.60|
> | Qwen2.5-3B |GPT-4o|50.67|92.67|82.67|36.00|69.33|66.27|
>
> These results indicate that BSD’s gains come from the decomposition strategy plus OT/OI-guided search, rather than from a particular hand-picked LLM. We will add a short ablation table reporting ASR and failure rates for different text-only decomposers in the appendix.

---

### Official Review · Reviewer_bfKP · 2025-11-07

**Soundness:** 2
**Presentation:** 2
**Contribution:** 2
**Rating:** 4
**Confidence:** 2

**Summary:**

This paper presents a systematic study of multimodal jailbreaks and introduces a quantitative framework (on-topicness, OOD-intensity, harmfulness, refusal) along with the Balanced Structural Decomposition (BSD) attack. BSD adaptively decomposes malicious prompts to balance semantic relevance and distributional novelty, achieving improvements in attack success.

**Strengths:**

- This paper defines and separates on-topicness and OOD-intensity, enabling structured, interpretable analysis of jailbreak mechanisms in multimodal LLMs.

- This paper proposes BSD, a recursive decomposition framework that operationalizes this balance and achieves improved attack success rates.

**Weaknesses:**

- The contribution is primarily heuristic and engineering-focused. BSD refines existing decomposition-based attacks rather than introducing new theoretical insights into model vulnerability.

- The BSD framework mainly extends prior decomposition-based attacks with additional heuristics (WBS, SDU) but lacks causal or mechanistic justification for why balancing on-topicness and OOD-intensity fundamentally increases vulnerability.

- The hierarchical search and threshold settings introduce many heuristic hyperparameters without sensitivity or stability analysis, making the method’s reproducibility and robustness across models and datasets uncertain.

**Questions:**

- How sensitive are the results to the embedding model choice and the specific OT/OI computation?

- Can the authors demonstrate that BSD captures intrinsic safety vulnerabilities rather than benchmark-specific weaknesses?

- Would the observed OT–OI balance remain effective under adaptive or adversarially retrained guard models?

---

> ### Author Response · Authors · 2025-11-30
> **Rebuttal (Part 1)**
>
> We thank the reviewer for the thoughtful and constructive feedback. Below we address the raised concerns point-by-point.
>
> ## Contribution is heuristic / lacks theoretical insight. [W1, W2, Q2]
>
> We respectfully clarify that our contribution goes beyond engineering refinement：
>
> Here we provide a mechanistic perspective on why balancing on-topicness (OT) and OOD-intensity (OI) specifically weakens safety alignment in multimodal LLMs.
>
> Recent work demonstrates that **LLMs encode harmfulness and refusal in separable token directions in embedding space** (*Zhao, Jiachen, et al. Llms encode harmfulness and refusal separately.*, NeurIPS 2025). That paper further argues that many existing jailbreak methods primarily work by suppressing the **refusal** signal, while leaving the model’s internal representation of **harmfulness** largely intact, which makes them easier to detect and defend against.
>
> In contrast, our mechanistic probing shows that BSD simultaneously reduces **both** directions:
> - significantly lower harmfulness signals in instruction-aligned tokens ($t_\text{inst}$), and
> - significantly lower refusal activation in post-instruction tokens ($t_\text{post-inst}$).
>
> This dual suppression explains why BSD’s balanced OT–OI structure is particularly effective and harder to defend against:
>
> - OT preserves the task relevance needed for the model to continue generating detailed responses,
> - OI injects semantics that systematically mislead harmfulness and refusal detectors,
> - Together they suppress both harmfulness perception and refusal activation while maintaining strong compliance signals.
>
> Thus, our method does not merely tune heuristics. It targets **two orthogonal safety-critical internal representations** and **extends prior findings beyond refusal-only suppression**, revealing a previously unobserved vulnerability in jailbreaking MLLM.
>
> We will add this mechanistic analysis to the revision, including diagnostic plots on  the extra page.
>
> ---
>
> ## Sensitivity analysis [W3]
>
> We have added a comprehensive sensitivity analysis on the HADES dataset covering recursive width $ W_\text{max} $ and tree depth $H_\text{max}$:
>
>
> ### Recursive width / depth ablation (ASR, higher is better)
>
> | $W_\text{max}$| $H_\text{max}$| Victim Model | Animal | Financial | Privacy | Self-Harm | Violence | Average |
> |------------|------------|--------------|-----|-----|-----|-----|-----|-----|
> | 7 (default) | 3 (default) | GPT-4.1      |43.33|88.67|78.67|28.00|64.67|60.67|
> | 7 (default) | 3 (default) | GPT-4o       |58.00|94.00|92.67|42.67|80.67|73.60|
> | 3          | 3 (default) | GPT-4.1      |49.33|82.67|67.33|26.00|57.33|56.53|
> | 3          | 3 (default) | GPT-4o       |32.67|93.33|79.33|34.67|70.67|62.13|
> | 5 | 5          | GPT-4.1      |33.33|90.67|84.00|30.67|70.67|61.87|
> | 5 | 5          | GPT-4o       |46.00|82.00|66.67|29.33|57.33|56.27|
>
> We also tested image generation hyperparameters (steps, guidance scale, style prompt of `FLUX.1-Schnell`):
>
> ### Visual cue parameter ablation (ASR, higher is better)
>
> | Guidance Scale | Steps        | Style          | Victim Model | Animal | Financial | Privacy | Self-Harm | Violence | Average |
> |------------|------------|--------------|-----|-----|-----|-----|-----|-----|-----|
> | 10 (default)   | 20 (default) | Anime (default) | GPT-4.1      |43.33|88.67|78.67|28.00|64.67|60.67|
> | 10 (default)   | 20 (default) | Anime (default) | GPT-4o       |58.00|94.00|92.67|42.67|80.67|73.60|
> | 4              | 20 (default) | Anime (default) | GPT-4.1      |43.33|79.33|62.00|25.33|58.00|53.60|
> | 4              | 20 (default) | Anime (default) | GPT-4o       |33.33|92.67|81.33|33.33|66.67|61.47|
> | 10 (default)   | 10           | Anime (default) | GPT-4.1      |50.00|86.00|68.67|32.00|64.67|60.27|
> | 10 (default)   | 10           | Anime (default) | GPT-4o       |27.33|85.33|82.00|24.67|62.00|56.27|
> | 4   | 10           | Anime (default) | GPT-4.1      |49.33|88.00|63.33|32.67|62.67|59.20|
> | 4   | 10           | Anime (default) | GPT-4o       |36.00|91.33|75.33|36.00|68.67|61.47
> | 10 (default)   | 20 (default) | Realistic       | GPT-4.1      |36.67|42.00|62.67|22.00|55.33|43.73
> | 10 (default)   | 20 (default) | Realistic       | GPT-4o       |37.33|66.67|79.33|28.67|62.00|54.80|
>
>
> Across all variations with anime-style images, BSD’s success rates remain consistently high and clearly above the CS-DJ baseline (40.67 for GPT-4.1 and 30.27 for GPT-4o). We observe a drop when switching to **realistic** images, which we hypothesize is because safety-aligned models are more sensitive to photorealistic content than to anime-style cues. Even in this setting, BSD remains comparable to or stronger than CS-DJ.
>
> Overall, these results show that BSD is **robust to recursive width/depth and visual hyperparameters**, and its gains over CS-DJ persist even under degraded configurations. We will include the full tables and additional comparisons in the revised paper.

---

> ### Author Response · Authors · 2025-11-30
> **Rebuttal (Part 2)**
>
> ## Results may be embedding-dependent. [Q1]
>
> To evaluate whether our findings depend on a specific embedding space, we conducted an embedding model ablation using two widely adopted SBERT encoders: **all-MiniLM-L6-v2** and **all-mpnet-base-v2**.
>
> Across both encoders, BSD maintains consistently strong ASR on HADES. While per-category scores fluctuate slightly, the overall averages remain high for both GPT-4.1 and GPT-4o, indicating that our attack is **not tied to a particular embedding model** and generalizes across standard semantic encoders.
>
> ### Embedding model ablation (ASR, higher is better)
>
> | Embedding Model | Victim Model | Animal | Financial | Privacy | Self-Harm | Violence | Average |
> |-----------------|--------------|-----|-----|-----|-----|-----|-----|
> | all-MiniLM-L6-v2 (default) | GPT-4.1 |43.33|88.67|78.67|28.00|64.67|60.67|
> | all-MiniLM-L6-v2 (default) | GPT-4o |58.00|94.00|92.67|42.67|80.67|73.60|
> | all-mpnet-base-v2       | GPT-4.1 |50.00|82.00|72.00|27.33|61.33|58.53|
> | all-mpnet-base-v2      | GPT-4o | 34.00|89.33|88.00|34.67|64.00|62.00|
>
>
> We will include these embedding ablation results, along with a brief discussion, in the revised appendix.
>
> ---
>
> ## Performance against guard models [Q3]
>
> We already evaluate BSD against a **strong** guard-model pipeline (GuardReasoner-VL) in Sec. 4.3, and we now further extend this analysis to include **LLaVA-Guard**:
>
>
> ### Guard model evaluation (ASR, higher is better)
>
> | Method | Guard Model            | Animal | Financial | Privacy | Self-Harm | Violence | Average |
> |------------------------|-----|-----|-----|-----|-----|-----|-----|
> | Ours | GuardReasoner-VL-3B |99.33|98.87|98.87|98.77|97.33|98.40|
> | CS-DJ | GuardReasoner-VL-3B |89.33|79.33|78.00|90.00|77.33|82.80|
> | Ours | GuardReasoner-VL-7B |89.33|80.67|84.00|86.00|78.00|83.60|
> | CS-DJ | GuardReasoner-VL-7B |79.33|60.67|49.33|61.33|78.00|65.73|
> | Ours | LLaVA-Guard-0.5B |100|100|100|100|100|100|
> | CS-DJ | LLaVA-Guard-0.5B |100|98.67|100|98.67|98.67|99.20|
> | Ours | LLaVA-Guard-7B |90.67|96.00|92.00|96.67|88.00|92.67|
> | CS-DJ | LLaVA-Guard-7B |87.33|98.33|98.67|88.00|86.00|89.87|
>
> Both our method and the CS-DJ baseline can achieve high ASR under LLaVA-Guard, indicating that current multimodal guard models remain vulnerable to carefully structured attacks. Importantly, **BSD consistently outperforms CS-DJ under the GuardReasoner-VL-3B/7B pipeline and also maintains a margin on LLaVA-Guard-7B**, showing that our gains are not tied to a single guard model.
>
> These results support our claim that BSD exploits **intrinsic vulnerabilities in cross-modal refusal grounding**, rather than focusing to a particular benchmark or guard configuration. We will add the full guard-model comparison tables and discussion to the appendix in the revised version.

---

### Author Response · Authors · 2025-12-03
**General Response by Authors**

Dear Area Chairs and Reviewers,

We sincerely thank you for the time, effort, and the constructive feedback on our submission.

Our general response is organized into three parts:
- Reviewers' recognition of our work
- Summary of our responses and paper updates
- Key Strengths of our work
## 1. Reviewers' recognition of our work
We are encouraged that all the reviewers expressed positive impressions of our work, including:
- **A clear and well-supported motivation:** We aim to an underexplored but practical limitation of existing OOD-style jailbreaks by foucusing on the fundamental **On-Topicness / OOD-Intensity trade-off** in multimodal LLM jailbreaks (bfKP, 16QD, FV5b, Hbdv).
- **A simple yet effective attack method:** Our proposed BSD explicitly balances OT and OI, achieving state-of-the-art performance on jailbreaking most advanced commercial and open-source models. (16QD, FV5b, Hbdv)
- **Extensive experiments and analysis:** Reviewers highlighted the breadth of our experiments across 13 multimodal LLMs (extended to 15 after rebuttal), two guard models (extended to four), and three widely-used benchmarks (HADES, MMSafetyBench, AdvBench-M), together with detailed qualitative and mechanism analysis (added during rebuttal), which together strengthen the persuasiveness of our claims (16QD, FV5b, Hbdv).
- **Clarity and presentation:** Reviewers highlighted that our paper is well-written and easy to follow, making the technical ideas and empirical insights accessible and convincing. (FV5b)
## 2. Summary of our response and paper updates

During the rebuttal, we provided **point-to-point** responses to all the weaknesses and questions and revised our paper accordingly. The main updates are summarized below:
- Explanation and refinement of our method
|Updates|Reviews|Locations|
|-|-|-|
|Mechanism insight| bfKP(W1, W2, Q2), Hbdv(W3)|Fig. 5, Sec. 4.6|
|Design of harmfulness score|FV5b(W3, W4, W5)|Sec 3.3|
|OI calculation reliability analysis|FV5b(W2)|Table 10, Appx C.6|

- Additional experiments
|Updates|Reviews|Locations|
|-|-|-|
|Additional baselines (JOOD)|FV5b (W6), Hbdv(W4, Q4)|Table 15, Appx C.10|
|Additional victim Models (GPT-5/5.1)|FV5b (W6)|Table 1, Sec 4.1, Sec 4.2, Appx B|
|Additional results on guard models|bfKP (Q3), 16QD(W3, Q4)|Table 2, Sec 4.3|

- **Additional ablation studies**
|Updates|Reviews|Locations|
|-|-|-|
|Sensitivity analysis on decomposition hyper-parameters $W_\text{max}$, $H_\text{max}$|bfKP(W3)|Table 6, Appx C.2|
|Decomposition model ablation|16QD(Q5), FV5b(W1), Hbdv(W1, Q1)|Table 7, Appx C.3|
|Visual cue hyper-parameter ablation|bfKP(W3), 16QD(W2, Q2, Q3), Hbdv(W2, Q2)|Table 9, Appx C.5|
|Visual component ablation|16QD(W2, Q2, Q3), Hbdv(W2, Q2)|Table 3, Sec 4.7|
|Embedding model ablation|bfKP (Q1)|Table 8, Appx C.4|
- **Paper writing**
|Updates|Reviews|Locations|
|-|-|-|
|Related work revision|16QD (W1, Q1), Hbdv (W4, Q4)|Sec 2.2|
|Visibility improvements|Hbdv|Fig 1, 8, 9, Line 256|

## 3. The strengths of our work

We summarize the main strengths of our work as follows:
- **A novel evaluation lens for jailbreaking MLLMs:** We propose a four-axis framework (Input: On-Topicness / OOD-Intensity, Output: Harmfulness, Refusal). **While most prior work relies on a single output-side metric,** we show that **measuring and exploit both input and output axes** is crucial for multimodal jailbreaks.
- **A simple yet effective attack method:** We use balanced OT-OI as a guiding heuristic and design a distraction-discription pipeline that **suppresses the model's internal belief of harmfulness and refusal**, leading to stronger jailbreaks and harder defenses while keeping the method simple and easy to implement.
- **Strong performance against advanced models:** We evaluate BSD on **15** powerful commercial and open-source models, including GPT-5.1, GPT-5, GPT-4o, GPT-4.1, Claude-sonnet-4, Gemini-2.5-pro, Qwen2.5-VL-32B, InternVL3-38B. BSD effectively jailbreaks these models and achieves large margins over baselines (**e.g. +65.73 on Gemini-2.5-pro, +43.33 on GPT-4o, +34.67 on GPT-5.1, +23.33 on Claude-sonnet-4**).
- **Comprehensive experimentation:** We conduct extensive experiments on three widely-used benchmarks (HADES, MM-SafetyBench, AdvBench-M), 15 victim models, and 4 guard models, covering 25 harmful topics, demonstrating both the broad effectiveness and robustness of our findings.
- **Generalisability and cost-effectiveness:** In contrast to methods that require tuning hyper-parameters for each dataset or model (e.g. JOOD and Andriushchenko et al. (*“Jailbreaking leading safety-aligned LLMs with simple adaptive attacks,”* ICLR 2025)), our method can **generate a single set of adversarial instructions that transfer across tasks and models**, achieving up to **225x lower API call** cost than JOOD.

Finally, we would like to thank the Area Chairs and all reviewers once again for their careful evaluation and constructive feedback on our work.

Best regards,
Authors of submission #7366

---

### Meta-Review · Area_Chair_igEn · 2025-12-23

**Summary:**

This paper focuses on the jailbreaking attacks against multimodal large language models (MLLMs). The contributions of this paper are twofold. (1) The paper introduces a four-axis evaluation framework that can more effectively identify truly effective jailbreaks. (2) The paper proposes a novel jailbreak method named Balanced Structural Decomposition (BSD). Experiments conducted on 15 commercial and open-source MLLMs demonstrate the effectiveness of BSD. Although the attack performance on closed-source models is impressive, several issues remain unaddressed. The primary experiments still rely on ASR for discussion and do not consistently employ the proposed four-axis evaluation framework for analysis, which somewhat deviates from the original intent of introducing this framework. Furthermore, the main experimental comparison is limited to two baselines (FigStep, CS-DJ). Lastly, the paper does not discuss any potential defense strategies.

**Reviewer Concerns:**

Addressed Concerns:
- Additional experimental results on guard models.
- More ablation studies.

Unaddressed Concerns:
- Conduct comparisons with more baseline attacks.
- Lack of new theoretical insights.

**Reviewer Scores:**

Unfortunately, I do not see any further discussion from the reviewers following the authors' response.

---

### Decision · Program_Chairs · 2026-01-26

Reject